# Performance of a Direct Methane Solid Oxide Fuel Cell Using Nickel-Ceria-Yttria Stabilized Zirconia as the Anode

**DOI:** 10.3390/ma13030599

**Published:** 2020-01-28

**Authors:** María José Escudero, María Pilar Yeste, Miguel Ángel Cauqui, Miguel Ángel Muñoz

**Affiliations:** 1Department of Energy, CIEMAT, 28040 Madrid, Spain; 2Department of Material Science, Metallurgical Engineering and Inorganic Chemistry, Faculty of Sciences, University of Cadiz, E-11510 Puerto Real, Cadiz, Spain; pili.yeste@uca.es (M.P.Y.); miguelangel.cauqui@uca.es (M.Á.C.); miguel.munoz@uca.es (M.Á.M.)

**Keywords:** alternative anode, Ni-Ce-YSZ, methane oxidation, SOFC, impedance spectroscopy, long-term tests

## Abstract

A nickel-ceria-yttria stabilized zirconia (Ni-CYSZ) cermet material was synthesized and tested as the anode for the direct oxidation of methane in a solid oxide fuel cell (SOFC) with YSZ as the electrolyte and strontium-doped lanthanum manganite (LSM) as the cathode. Initially, the electrochemical behavior was investigated under several load demands in wet (3% H_2_O) CH_4_ at 850 °C during 144 h using I-V curves, impedance spectra, and potentiostatic measurements. Long-term tests were subsequently conducted under 180 mA·cm^–2^ in wet CH_4_ for 236 h and dry CH_4_ for 526 h at 850 °C in order to assess the cell stability. Material analysis was carried out by SEM-EDS after operation was complete. Similar cell performance was observed with wet (3% H_2_O) and dry CH_4_, and this indicates that the presence of water is not relevant under the applied load demand. Impedance spectra of the cell showed that at least three processes govern the direct electrochemical oxidation of methane on the Ni-CYSZ anode and these are related to charge transfer at high frequency, the adsorption/desorption of charged species at medium frequency and the non-charge transfer processes at low frequency. The cell was operated for more than 900 h in CH_4_ and 806 h under load demand, with a low degradation rate of ~0.2 mV·h^–1^ observed during this period. The low degradation in performance was mainly caused by the increase in charge transfer resistance, which can be attributed to carbon deposition on the anode causing a reduction in the number of active centers. Carbon deposits were detected mostly on the surface of Ni particles but not near the anode/electrolyte interface or the cerium surface. Therefore, the incorporation of cerium in the anode structure could improve the cell lifetime by reducing carbon formation.

## 1. Introduction

Solid oxide fuel cells (SOFCs) are highly efficient energy conversion devices with the characteristics of fuel flexibility and low pollutant emissions [1]. In addition to pure hydrogen, SOFCs can theoretically be operated with hydrocarbon fuels such as methane (the main component of natural gas). The direct use of methane eliminates the need for pre-reformers, thus reducing the complexity, size, and cost of the overall SOFC system [2]. However, the incompatibility of the Ni/yttrium-stabilized zirconia (YSZ) cermet, the most widely used anode material in SOFC, with hydrocarbon fuels makes it difficult to exploit the great benefits because nickel catalyses the decomposition of hydrocarbon fuels, which results in a severe deposition of carbon at the anode [3]. The carbon mainly forms three structures: amorphous, filamentous, and graphitic [4]. Taking into account the high operating temperature of SOFCs (600–1000 °C), the graphitic carbon is the most favored structure, although amorphous carbon and filamentous carbon have also been observed [4,5,6,7]. The amorphous carbon is formed preferentially up to 600 °C, but it can grow into filamentous carbon and ultimately be converted to graphitic forms at high temperature. The presence of graphite formed by carbon that has diffused through the anode causes the disintegration of Ni metal to form a powder in a process that is known as “metal dusting”. The presence of carbon fibers at high temperature (>600 °C) is caused by the dissolution of adsorbed carbon in the metal crystallite, its diffusion through the metal, and its precipitation at the back of the metal particle. A priori, filamentous carbon is easier to remove by electrochemical oxidation than graphitic carbon because the former can be combusted/oxidized at lower temperatures. However, carbon fibers can cover the active sites of Ni-YSZ and become incorporated into the anode lattice, thus provoking structural deformation and fragmentation that causes more significant performance degradation than graphite [5,7]. The accumulation of carbon on Ni therefore causes unwanted complications such as blocking of active sites on the anode, loss of nickel catalyst, and loss of structural integrity [8].

Even with the disadvantages outlined above, Ni-based anodes are still attractive not only because of their unequalled fundamental characteristics, including outstanding catalytic performance, high electronic conductivity, and good chemical stability in relation to the electrolyte, but also due to their relative ease of manufacture, appropriate mechanical strength, and low cost [9,10]. Several strategies can be adopted to improve the performance and coking resistance in hydrocarbon-fuelled SOFCs. One effective approach is the use of nickel-based anodes combined with materials with excellent catalytic activity and coke resistance, such as ceria.

CeO_2_-based mixed oxides have received attention as components of anode materials for SOFCs.These compounds present mixed ionic and electronic conductivity in reducing atmospheres, excellent catalytic activity for the combustion of hydrocarbons, and an adequate resistance to carbon deposition [11]. It has also been reported that the addition or coating of CeO_2_ increases the methane oxidation activity and reduces the carbon deposition in the nickel Ni/YSZ. This behavior is attributed to ceria possibly enlarging the triple phase boundary (TPB) due to its mixed ionic-electronic conductivity and increasing the rate of oxidation of carbon components in active regions by supplying extra oxygen ionic flux. During this process, the valence Ce(IV) diminishes, thus producing an increase of oxygen vacancies in CeO_2−x_ [12,13,14]. These properties suggest the potential application of these materials as SOFC anodes for the direct electrochemical oxidation of hydrocarbons. Nevertheless, the main weakness of ceria is its low thermal stability under SOFC preparation and operation conditions because the ceria surface area is strongly affected by calcination temperature [15]. He at al. [16] reported that the area specific resistance of ceria increased by around a factor of three when the ceria was calcined at 1250 °C instead of 550 °C, with this increase being due to a loss of ceria surface produced by particle sintering. An effective attempt to improve the thermal stability and carbon tolerance of this material for SOFC anode applications was made by replacing pure ceria with ceria-zirconia mixed oxides. It has been widely reported that the incorporation of Zr^4+^ in the lattice of CeO_2_ enhances its reducibility and oxygen storage capacity as well as the thermal stability and electrical conductivity of its fluorite structure [17]. The structural and redox properties of ceria-zirconia mixed oxides depend on their composition, and they are markedly influenced by the synthesis procedure and the redox and/or thermal treatments performed. 

In a previous publication, we reported that the deposition of ceria on the surface of zirconia and yttria-doped zirconia nanocrystals benefits the structural and chemical interactions that can occur between these components using specific high temperature redox ageing treatments [18]. These materials exceed the performance of noble metal loaded catalysts based on bulk ceria-zirconia mixed oxides in terms of reducibility and oxygen exchange capacity. Therefore, these compounds have attractive properties to be investigated as anode materials for SOFCs directly fed with hydrocarbons because they could enhance the thermal stability and carbon tolerance. 

In the study reported here, an alternative anode material for direct methane SOFC based on ceria-yttria-doped zirconia nanocrystals loaded with nickel (Ni-CYSZ) has been synthesized by a novel route developed by our group. The addition of ceria to the structure could act by minimizing the carbon formation, which would lead to stabilization of the anode structure. The anode material was tested in a single cell with 8 mol% yttria stabilized zirconia (YSZ) as the electrolyte and (La_0.80_Sr_0.20_)_0.95_MnO_3−δ_ (LSM) as the cathode. The performance and long-term stability of the cell for direct methane oxidation (wet and dry) were investigated using direct current (DC) and alterning current (AC) electrochemical techniques at 850 °C. In addition, scanning electron microscopy (SEM) and energy dispersive spectroscopy (EDS) were employed to study the morphology of the anode after testing.

## 2. Materials and Methods

### 2.1. Synthesis and Characterization of Anode Material

Ni-CYSZ powders were synthesized using a previously reported method [18,19]. The CYSZ support was prepared by incipient wetness impregnation of a commercial yttria-stabilized zirconia (YSZ, with 15 mol% of Y provided by Tecnan-Nanomat S.L., Navarra, Spain) with an aqueous solution of Ce(NO_3_)_3_·6H_2_O. The sample with a cerium load of 15 mol% was dried at 110 °C and then calcined at 500 °C for 1 h. Prior to Ni incorporation, the calcined support was treated using a specific redox-ageing procedure (SRMO protocol) which involves a Severe Reduction (SR) followed by a Mild Oxidation (MO) treatment, as described in [18]. Briefly, in this treatment, the sample was heated at 950 °C in H_2_ (5%)/Ar for 2 h. The gas flow was then changed to He for 1 h and the sample was cooled down to room temperature. The last step consisted of a re-oxidation with O_2_ (5%)/He, first at room temperature and subsequently at 500 °C for 1 h. The nickel catalyst was obtained by the wet-impregnation method, using an aqueous solution of Ni(NO_3_)_2_·6H_2_O as precursor. Four impregnation cycles were carried out to achieve a nickel loading of 11 mol%. The powders were calcined at 500 °C for 1 h after each impregnation.

The structure of Ni-CYSZ was studied at room temperature by X-ray diffraction (XRD) using a Bruker diffractometer model D8 ADVANCE (Bruker Corporation, Billerica, MA, USA) employing Cu-K_α_ as the radiation source. X-ray diffractograms were collected in the 2θ range from 3 to 75° with 0.01 step and 3 s counting time per point.

### 2.2. Single Cell Fabrication and Test

An electrolyte supported cell with an active area of 0.25 cm^2^ and 8 mol% yttria stabilized zirconia (YSZ, Pi-Kem) as electrolyte was prepared using a fabrication method based on a previous report [20]. A pellet of YSZ was fabricated from the commercial powder by uniaxial pressure and calcination at 1500 °C for 10 h and its thickness was reduced by mechanical grinding. After sintering, the dense YSZ pellet had a diameter of ~11 mm and a thickness of ~0.4 mm. A porous YSZ layer was deposited onto the anode side of the electrolyte pellet by tape casting using polymethylmethacrylate (PMMA) as a pore former and then calcined in air at 1400 °C for 2 h. This buffer layer of YSZ improves the adhesion of anode ink on the electrolyte. The anode ink was prepared by mixing the anode powders with a suitable amount of binder (Decoflux, Zschimmer and Schwartz) to obtain a slurry that was screen-printed onto the porous YSZ and then heated at 850 °C for 2 h. The lanthanum strontium manganite (LSM, NextEnergy, Milan, Italy) cathode ink was also applied by screen-printing onto the other side of the electrolyte and then calcined at 850 °C for 2 h. 

Platinum and gold pastes were painted onto the cathode side and anode side, respectively, for use as current collectors and the cell was heated at 850 °C for 1 h. Finally, the cell was fixed to an alumina tube with a ceramic seal (Aremco, Ceramabond 552, Valley Cottage, NY, USA) and positioned in a furnace. 

The cell was heated up to 900 °C under wet (3% H_2_O) H_2_ and held for an hour in order to reduce the metal oxide. The cell was then cooled down to 850 °C. The cell was characterized using both wet (~3 vol.% H_2_O) and dry CH_4_ as fuel with a constant flow rate of 50 mL·min^–1^ while the cathode was exposed to ambient air at 850 °C. Long-term tests were carried out at a constant current density of 180 mA·cm^–2^. The current voltages and impedance spectra were measured intermittently. An Autolab system (PGSTAT30 and FRA2 module, Eco Chemie, Kanaalweg, the Netherlands) was used for the electrochemical measurements. The I-V curves were obtained using linear sweep voltammetry at a scan rate of 10 mV·s^−1^ and the impedance spectra were obtained at open circuit in a frequency range of 50 kHz–10 mHz with a current amplitude of 5 mA.

### 2.3. Post-Test Analysis

After testing, the microstructure and composition of the cell were examined by scanning electron microscopy (SEM) with energy dispersive spectroscopy (EDS) and mapping techniques using an FEI Nova NanoSEM 450 microscope (Hillsboro, Oregon, USA) with different acquisition modes.

## 3. Results and Discussion

### 3.1. Structural Characterization of the Anode Material

The XRD pattern of the Ni-CYSZ material calcined at 500 °C is shown in Figure 1. The diffractogram contains peaks corresponding to the tetragonal structure of the YSZ support (JCPDS 30-1468) and weak peaks at 2θ = 37.3° and 46.7° that can be ascribed to the cubic structure of NiO (JCPDS 65-6920). The absence of peaks associated with pure ceria suggests that after SRMO activation, ceria is incorporated at surface level in the YSZ structure, as demonstrated in a previous study [18]. Average crystallite sizes were estimated by the Scherrer equation, with values of 20.8 nm for the CYSZ support and 17.1 nm for NiO.

### 3.2. Electrochemical Performance and Long-Term Stability under Dry and Wet CH_4_

The operating temperature was fixed at 850 °C because the LSM cathode usually operates at high temperature (800–1000 °C) due to its notable overpotential enhancement below 800 °C, which is caused by the relatively low reaction rate for oxygen reduction [21]. It was also expected that the high thickness of the supported electrolyte (~400 μm) could reduce the cell performance. The open circuit voltage (OCV) of the cell was measured during 1 h under wet (3% H_2_O) and dry CH_4_ at 850 °C. The OCV values remained perfectly stable with both wet and dry methane, which suggests that the cell is stable under both sets of operating conditions. This finding could indicate the absence of carbon deposition. The OCV of the cell was around 0.93 and 0.96 V under wet (3% H_2_O) and dry CH_4_, respectively. The slightly lower OCV value under wet CH_4_ could be directly related to the presence of water increasing the partial pressure of oxygen at the anode.

It is well known that the theoretical OCV is a thermodynamic parameter that depends on the oxygen partial pressure of the cathode and anode, with this parameter being independent of the reaction kinetics. However, in practice, significant oxygen exchange must occur between the anode and the fuel gas and this could affect the OCV value. Given the high operating temperature (850 °C), methane could decompose in an SOFC not only at the three-phase boundary (TPB) but also in the whole anode region. When dry or wet methane is fed to the SOFC anode, H_2_, CO, CO_2_, and H_2_O could be formed by partial electrochemical oxidation of CH_4_, direct electrochemical oxidation of CH_4_, and steam reforming of CH_4_. In addition, deposited carbon can be obtained by methane decomposition on Ni-based anodes [22]. Various chemical and electrochemical reactions can occur between the different reagents and products, as well as with the O^2–^ from the cathode side. The likely electrochemical reactions are detailed below [23]:CH_4_ + 4O^2−^ → CO_2_ + 2H_2_O + 8e^−^,(1)
CH_4_ + O^2−^ → CO + 2H_2_ + 2e^−^,(2)
H_2_ + O^2−^ → H_2_O + 2e^−^,(3)
CO + O^2−^ → CO_2_ + 2e^−^,(4)
C + O^2−^ → CO + 2e^−^.(5)

Moreover, the possible chemical reactions are:CH_4_→ C + 2H_2_,(6)
CH_4_ + H_2_O → CO + 3H_2_,(7)
CO + H_2_O → CO_2_ + H_2_,(8)
CH_4_ + CO_2_ → 2CO + 2H_2_,(9)
2CO → C + CO_2_,(10)
C + H_2_O → CO + H_2_.(11)

Both experimental OCV values are lower than the theoretical values for complete oxidation of methane (~1.05 V) and partial oxidation of methane (~1.25 V) [24]. This indicates the presence of partial oxidation products and chemical reactions at the anode region. Therefore, it is possible that the different OCVs that were observed are due to the contribution of each reaction (electrochemical and chemical) to different extents. The contribution of each reaction depends mainly on the anode material, microstructure, the cell configuration, and the operation conditions.

Low OCV values in wet and dry methane have also been reported for electrolyte-supported cells by other authors. Buccheri et al. [25] reported an OCV of 0.92 V in dry CH_4_ at 750 °C with Ni-YSZ. Eguchi et al. [26] also obtained lower OCV values for an electrolyte-supported cell with Ni-YSZ operated under various CH_4_ concentrations after internal reforming of CH_4_-H_2_O gas mixtures at 1000 °C. Additionally, Gorte et al. [27] found an OCV of 0.9 V for dry CH_4_ at 700 °C using an anode supported cell with Cu-CeO_2_. They also interpreted the low value as being due to the fact that hydrocarbons can be oxidized in multiple steps, so that the equilibrium is established between the chemical fractions and partial oxidation products.

In addition to an optimum electrocatalytic activity for methane oxidation, an appropriate performance stability that indicates the inhibition of carbon deposition is also required. Current density is a crucial operating parameter to remove carbon deposits on the anode. Therefore, the cell was initially tested under wet CH_4_ (3% H_2_O) at 850 °C during 144 h by fixing different current density values, each for 24 h. The corresponding values of power density of the cell are displayed in Figure 2a. The current density load was fixed at 100, 120, 140, 160, 180, and 200 mA·cm^−2^ for 24 h. An increase in the cell performance with current density was observed, except for a load demand of 200 mA·cm^−2^. In general, a continuous increase in power density can also be observed with operating time at each load demand. On applying current to the cell, oxygen ions were supplied from the cathode side to the anode side through the electrolyte, and they reacted with deposited carbon formed by methane decomposition (Equation (6)) at the reaction zone (TPB), thus causing a refreshment of electroactive sites (mainly Ni). This process could produce a gradual increase in the cell performance with time at a given current density. After the refreshment of active sites, the oxidation of H_2_ (Equation (3)) and CO (Equation (4)) are the main reactions that take place. As the current increases, the flux of oxygen ions from the cathode to the anode increased, and these oxygen species may mostly be employed for direct oxidation of CH_4_, CO and H_2_. However, when the direct oxidation of methane occurs (Equation (1)), a proportion of methane still participates in partial oxidation (Equation (2)). 

The partial oxidation of methane could be more predominant at lower current density while a direct oxidation of CH_4_ predominates with increasing current density. On the other hand, the production rate of H_2_O and CO_2_ is also enhanced as current intensity increases. Both CO_2_ and H_2_O are weak oxidizing agents that could oxidize Ni to NiO, thus reducing the catalytic activity of the anode. Therefore, the lower power density values observed with a load demand of 200 mA·cm^−2^ in comparison with that of 180 mA·cm^−2^ could be caused by higher formation of CO_2_ and H_2_O in the anode chamber increasing polarization losses. In addition, the excess of methane can give rise to surplus carbon deposition and this cannot be removed by electrochemical oxidation. Several authors also reported that the electrochemical reactions of methane at different current densities are governed by multiple factors such as gas composition and temperature [24,28,29,30,31]. In general, it was observed that, at low current density (low oxygen stoichiometry), the anode reaction is dominated by methane cracking and/or partial oxidation of methane, while, at higher current density (higher oxygen stoichiometry), the direct oxidation of methane reaction prevails.

It is worth noting the power density oscillations observed for the methane reaction in wet CH_4_ under all current density demands. These oscillations indicate a variation in the partial pressure of oxygen in the cell. Wang et al. [32] studied potential oscillations in the methane reaction on anodes of SOFC, and they reported that the potential oscillation is most likely caused by the adsorption and desorption of oxygen species on the active centers of the anode. Therefore, the increased magnitude of the power density oscillation with increasing load demand may be due to the enhancement of adsorbed/desorption oxygen species on the anode surface, as well as the possible cyclic processes of oxidation/reduction of Ni to NiO due to the presence of H_2_O and CO_2_. 

The I-V curves (Figure 2b) and impedance spectra (Figure 2c) of the cell were also recorded initially and after each load demand (24 h) in order to gain more information about the cell operation. 

The cell potential and power density of the cell under wet CH_4_ as a function of current density and time at 850 °C are shown in Figure 2b. A significant enhancement of the values of voltage, power density, and current density was observed after load demands in comparison with the values measured initially (i.e., before load demand). Both the maximum power density and maximum current density (V = 0 mV) of the cell increased up to ~95% and 88%, respectively, after 24 h under load. After that time, both variables continued to rise from 49 to 79 mW·cm^−2^ and from 194 to 294 mA·cm^−2^, respectively, up to 120 h with increasing time after a load demands from 100 to 180 mA·cm^−2^. At 144 h, a reduction of power density could be observed after load demand of 200 mA·cm^−2^. The I-V curves were measured consecutively in each load demand test and, as a result, the variation in the cell performance could be associated with the change in carbon deposits on the active centers. The results therefore indicate the relevance of the electrochemical reactions under load demand.

The impedance spectra were measured under OCV conditions in wet CH_4_ at 850 °C after each I-V curve and as a function of time (Figure 2c). Three arcs can be clearly differentiated in all Nyquist diagrams in the high frequency (HF), medium frequency (MF), and low frequency (LF) regions. Both HF and MF arcs decreased significantly with increasing operating time, whereas the LF arc appeared to change less with operating time. The impedance spectra shown in Figure 2c were fitted to an equivalent circuit, LR_Ω_(R_HF_Q_HF_)(R_MF_Q_MF_)(R_LF_Q_LF_) (shown in the inset of Figure 2c), using the Zview program. L is the inductance element produced by the electrochemical device and the connection cables. R_Ω_ represents the ohmic resistance of the cell (mostly attributed to the electrolyte) and the (RQ) components correspond to those involved in electrode processes. In this notation, R is a resistance and Q is a constant phase element. The capacitance and relaxation frequency of each contribution can be obtained by Equations (12) and (13), respectively [33]:(12)ϖi=RQ−1ni,
(13)C=R1−nQ1n.

The polarization resistance (R_p_) of the cell is determined by R_HF_ + R_MF_ + R_LF_ and contains the resistances of electrochemical and non-electrochemical processes from both the anode and cathode. Good adjustments between the equivalent and the experimental circuits were observed. The resistance and capacitance values obtained from fitting to the impedance data for different operating times are provided in Table 1. It is worth noting that the inductance values are excluded because they are not characteristic of the studied cell. 

In a previous work [20], the electrochemical operation of an SOFC with Mo-Ni-Ce as the anode material was investigated in H_2_, CH_4_, and H_2_/CH_4_ mixtures at 750, 800, and 850 °C with static air as the oxidant. The impedance spectra were also fitted to the same equivalent circuit and based on the dependence of resistances (R_HF_, R_MF_ and R_LF_) and capacitances (C_HF_, C_MF_ and C_LF_) with the gas composition and temperatures, as well as the range of relaxation frequencies and capacitances, with each arc attributed to a different process. Thus, the HF arc was assigned to electrochemical charge transfer at the TPB, the MF arc to adsorption or desorption of reactant species on the electrodes (anode and cathode), and the LF arc with the oxygen exchange reaction and/or diffusion reactions on the electrode and possible chemical reactions (non-charge processes). Similar results were obtained by our group on using a single cell with Cu-Ni-CeO_2_ as the anode material tested in H_2_ and CH_4_ at 750 and 800 °C [22]. Furthermore, this equivalent circuit was also used by other authors [34,35] to study methane oxidation in SOFCs.

In this work, the characteristic relaxation frequencies obtained from the equivalent circuit fitting are in the range of 6.2–3.0 KHz for the HF arc, 14–10 Hz for the MF arc and 0.052–0.048 Hz for the LF arc, and the C_HF,_ C_MF_, and C_LF_ values are in the range 9–36 μF·cm^−2^, 5–11 mF·cm^−2^ and 4.6–6.1 F·cm^–2^, respectively. These characteristic relaxation frequencies and capacitances are of the same order of magnitude as those obtained in the previous works [20,22] under CH_4_. Thus, the HF capacitance value is the usual value for a double-layer capacitance (~10 μF) [36,37]. The HF arc process is mostly assigned to the electrochemical charge transfer at the triple phase boundary (TPB). The MF capacitance value is on the order of 10 mF·cm^–2^, and this is typical for an adsorption/desorption process of a charged species on the electrode surface [38,39]. On the other hand, the LF capacitance value is significantly higher, and it is therefore not related to interface capacitance or adsorption on the surface; in fact, a bulk process should occur to explain this high value. Ni-CYSZ is a mixed ionic conductor, and it can vary the oxygen stoichiometry if the electrode potential is changed. In this case, the large chemical capacitance could be caused by the oxygen incorporation/release reaction at the electrodes [40,41]. The LF arc could then be related with non-charge transfer processes that include the oxygen exchange reaction and/or diffusion reactions on the anode and possible chemical reactions. Jiang et al. [35] also reported on the electrocatalytic activity of (La_0_._75_Sr_0_._25_)(Cr_0.3_Mn_0.5_)O_3_/YSZ composites for the methane oxidation reaction using three distinguishable arcs. They also related the LF arc (0.025 Hz at 900 °C) with the non-charge transfer processes. Large capacitances associated with the LF arc were also found by Chen et al. [42] in Ni-SDC anodes in H_2_. They attributed these values to the dissociation/diffusion processes of hydrogen on the Ni surface, which are favored by oxygen species. As the LSM cathode is exposed to static air during testing at the same temperature, 850 °C, the variation of the polarization resistances is mainly related to modifications in the anode material.

R_Ω_ decreased with increasing operating time, but this change became less significant after 96 h. The values of R_HF_ and R_MF_ showed similar behavior. Both resistance values decreased with increasing operating time up to 120 h, after which an increase in resistance values was detected when the cell was tested under a load demand of 200 mA·cm^–2^. Note that the decrease in R_HF_ is more marked than that of R_MF_ and this is more notable in the first 48 h (Table 1). This trend could be caused by the electrochemical oxidation of deposited carbon formed by methane decomposition (Equation (6)) on the nickel surface, which would increase the number of active centers that favor charge-transfer processes. 

R_LF_ showed minor changes and an opposite trend with operating time when compared to R_HF_ and R_MF_. The R_LF_ values increased with operating time up to 120 h and then decreased. 

The decrease of R_Ω_ and R_HF_ assigned to charge transfer process could be related to the moderate formation of carbon deposits under the operation conditions, with these deposits acting as bridges between Ni particles. The deposits not only enhance the electronic conductivity, but they also connect isolated regions. Similar behavior was observed by other authors [43,44,45], and this was explained by a model proposed by Gorte and co-workers [46]. They reported that some metal particles are not linked to the outside circuit and cannot contribute to the removal of electrons. In such a case, the whole region of isolated metal particles is useless for the electrochemical reaction. With the formation of a moderate amount of carbon deposits, the isolated metal particles could become electronically connected to the outside circuit. As a result, more of the anode surface is now involved in the electrochemical reaction, and an apparent decrease in the resistance values is observed. According to this model, the formation of small conductive carbonaceous species on the pores increases both the electronic conduction, thus decreasing R_Ω_, and the anode surface involved in the electrochemical reaction, thus reducing R_HF_. Moreover, the decrease of R_MF_ could also be associated with the increase in anode surface due to the adsorption of charged species, albeit to a minor extent. Nevertheless, the decreases in R_HF_ and R_MF_ could also be due to a slow reduction in the deposited carbon during the operation under load demand as a consequence of the reactions outlined above (Equations (5) and (11)). On the other hand, the increase of R_HF_ and R_MF_ observed at 144 h could be caused by an increase in carbon deposits, which begin to cover the active sites of the anode and reduce the active area for electrochemical reactions. However, we believe that the carbon deposits are still not sufficient to affect the electrical conductivity, and the R_Ω_ value is therefore not influenced. Lastly, the minor changes in R_LF_ could be related to the variation of oxygen stoichiometry when the electrode potential changes and/or non-charge chemical process with time, as mentioned above. It will be necessary to investigate the effect of fuel composition and temperature in order to clarify this point. 

The long-term stability of the cell with Ni-CYSZ was investigated under a load demand of 180 mA·cm^–2^ in wet (3% H_2_O) and dry CH_4_ at 850 °C during 236 h and 526 h, respectively (Figure 3). Significant power density fluctuations were observed and these revealed a change in the partial pressure of oxygen in the cell. As mentioned previously, such a phenomenon is most likely due to the adsorbed/desorbed oxygen species on the anode surface during the CH_4_ oxidation reaction as well as the possible cyclic processes of oxidation/reduction of Ni to NiO and vice versa. It should be noted that the power densities decreased slowly with operating time. In wet methane, the cell showed a slight increase in power density during the first 70 h, after which the power output decayed slowly with approximately 17% of the initial power density after testing for 236 h (a degradation rate of 0.29mV·h^−1^ or 0.05 mW·h^−1^ for 236 h). In dry CH_4_, the cell performance drops were around 21% and 41% after operating for 236 h and 526 h, respectively (a degradation rate of 0.38 mV·h^−1^ and 0.32 mV·h^−1^ or 0.08 mW·h^−1^ and 0.06 mW·h^−1^ for 236 h and 526 h, respectively). In general, these results revealed similar degradation rates in dry and wet CH_4_ during 236 h. However, it is clear that, after this time, the cell continues to degrade.

During the operating period in CH_4_ (wet and dry), I-V curves and impedance spectra (under open circuit conditions) were measured in order to obtain more information about the performance degradation of the cell. The I-V curves and Nyquist diagrams measured before and after the stability test (236 h) under wet (3% H_2_O) CH_4_ are shown in Figure 4. It can be seen that the cell reached a maximum power density of around 80 mW·cm^−2^, and the power density values did not change significantly at current densities below those corresponding to maximum power density, but a decrease in the maximum current density (V = 0 mV) from 297 to 257 mA·cm^-2^ was observed after operating under a load demand of 180 mA·cm^-2^ for 236 h (Figure 4a). The performance at higher current densities decreased after testing, while the Nyquist diagrams (Figure 4b) revealed an increase in the polarization resistance of ~15%, i.e., from 3.35 to 3.85 Ω·cm^2^.

In an effort to clarify the influence of wet or dry CH_4_ on the cell performance, the cell was fed with pure H_2_ before and after long-term testing in wet CH_4_ (3% H_2_O). Significant changes were not observed in the I-V curves measured (Figure 5) in H_2_ before and after the cell was operated with wet CH_4_, thus indicating that the poisoning by carbon deposition on the Ni surface is reversible. The cell was then operated in dry CH_4_.

The I-V curves and impedance spectra collected in dry CH_4_ before and after operating for 236 and 526 h under load demand are displayed in Figure 6. The initial OCV was 0.95 V and this decreased to 0.94 and 0.93 V after 236 and 526 h, respectively. In addition, both the peak power density and maximum current density diminished with operating time. Initially, the maximum power density was 80 mW·cm^−2^, and this dropped by ~14% and ~25% after operating for 236 h and 526 h, respectively; the maximum current diminished from 279 mA·cm^−2^ to 255 mA·cm^−2^ for 236 h and 238 mA·cm^−2^ for 526 h, which represents decreases of 9% and ~15%, respectively. 

It is worth emphasizing that the I-V curves and impedance spectra measured in dry CH_4_ at the beginning and after 234 h under dry CH_4_ (Figure 6) are very similar to those obtained in wet CH_4_ (3% H_2_O) (Figure 4). We believe that the presence of water in the fuel feed is not significant because water is produced when the cell is operated under load demand.

The Nyquist plots revealed that ohmic resistance increased from 0.50 Ω·cm^2^ to 0.58 Ω·cm^2^ for 236 h and 0.64 Ω·cm^2^ for 526 h, while the polarization resistance was enhanced by ~17% (from 3.42 to 3.99 Ω·cm^2^) and ~19% (from 3.42 to 4.10 Ω·cm^2^) after operating for 236 h and 526 h, respectively. The slight increase in R_Ω_ was not observed when the cell was operated under wet CH_4_, and this may be associated with the increase of carbon deposits on the anode surface with operating time. 

It is worth noting that all impedance spectra measured in dry CH_4_ also contained three arcs that are observed at high, medium, and low frequencies (Figure 4b,6b). This indicates that the electrochemical reactions in the cell (anode and cathode) are limited by at least three electrode steps. These impedance data were fitted to the equivalent circuit shown in Figure 2c; good agreement was observed between the experimental data and the calculated data. The capacitance values obtained for the high frequency arc (C_HF_), medium frequency arc (C_MF_), and low frequency arc (C_LF_) were in the range 44–26 μF·cm^−2^, 15–7 mF·cm^−2^, and 2–0.9 mF·cm^−2^, respectively. These results are of the same order of magnitude as the values reported in Table 1. The HF arc, MF arc, and LF are also assigned to electrochemical reactions, adsorption/desorption of charged species or surface diffusion of the adsorbed species, and non-charge transfer processes such as oxygen surface exchange, gas diffusion and possible chemical reactions, respectively. A comparison of R_Ω_ values (ohmic resistances) and R_HF_, R_MF_, and R_LF_ values (non-ohmic resistances) obtained by fitting the impedance data are displayed in Figure 7.

In wet CH_4_ (3% H_2_O) (Figure 7a), significant changes were not observed in the ohmic resistance (R_Ω_), but the polarization resistance (R_p_) increased by 15.6% from 3.33 to 3.85 Ω·cm^2^. Initially, R_p_ represents 86.5% of the total resistance but after testing for 326 h its contribution increased to 88.5%. The data fitting showed that the contributions of R_HF_, R_MF_ and R_LF_ to the polarization resistance were 43%, 36%, and 21% initially and 30%, 41%, and 20% after 236 h, respectively. The cell degradation could be related to an enhancement of the adsorption of charged species on the anode surface. This could in turn diminish the number of active centers increasing R_HF_, which is associated with electrochemical processes. In addition, non-charge transfer processes such as oxygen surface oxygen surface exchange, gas diffusion, and possible chemical reactions [41] related to R_LF_ could also be hindered.

In dry CH_4_ (Figure 7b), it was observed that R_Ω_ increased by 8% and 20% after testing under load demand for 236 h and 526 h, respectively, while R_p_ increased by 17% and 20%. Nevertheless, R_p_ represents the main contribution to the total resistance (R_t_), i.e., ~88%. The contributions of R_HF_, R_MF_ and R_LF_ to R_p_ were 34%, 45%, and 21% initially, but increased to 38%, 44%, and 17% for 236 h and 46%, 37%, and 17% for 523 h, respectively. In general, these values are similar to those obtained in wet CH_4_ in the same times (initially and after 236 h). The presence of 3% H_2_O in the feed is therefore not relevant when the cell is operated under a load demand of 180 mA·cm^-2^. For operating times higher than 236 h, the loss of electrochemical performance is mainly caused by the increase in R_HF_. This can be explained by the increase in the amount of carbon deposits with operating time, with these deposits covering the anode surface and blocking the anode pores, thus reducing their connectivity. This process could hinder the electrochemical reactions because these active sites lack fuel. Moreover, a proportion of the deposited carbon that is not involved in the electrochemical reactions is converted into residual carbon, which at higher levels may increase the ohmic resistance. However, we believe that this behavior is produced mainly by the operating time rather than the absence of water.

It is widely acknowledged that the formation of carbon in CH_4_-fuelled SOFCs mainly takes place via two reactions, pyrolysis (Equation (6)), and the disproportionation of CO (Equation (10)), the so-called Boudouard reaction, which is thermodynamically favored at temperatures below 700 °C. At temperatures above 700 °C, carbon deposits are mainly produced by the pyrolysis reaction [47]. It is evident that the complete oxidation of methane (Equation (1)) was not achieved. According to the operating temperature (850 °C), carbon formation on Ni-CYSZ during the long-term stability test principally occurred by the pyrolysis reaction (methane decomposition). The increase in carbon deposits produced by the pyrolysis reaction (Equation (6)) could lead to blockage of a higher number of active centers, and this would impede the electrochemical reactions.

It is difficult to compare these results with the literature data for SOFC with Ni-YSZ as anode in CH_4_ because of the different cell configurations (anode supported or electrolyte supported), microstructure of the electrodes, thickness of electrolyte, and fabrication methods for the cell and operating conditions. Nevertheless, electrolyte supported YSZ (~100 μm of thickness) using Ni-YSZ as the anode material and LSM as the cathode material gave a maximum power density of ~56 mW·cm^–2^ in wet CH_4_ at 900 °C [48] and 40 mW·cm^–2^ in dry CH_4_ at 800 °C [49]. Therefore, our cell showed higher performance despite the thickness of the electrolyte being higher (~400 μm).

After testing in dry CH_4_, the cell was operated in pure H_2_ and it reached a maximum power density of 50 mW·cm^−2^. This value is lower than obtained previously and after testing in wet CH_4_ (Figure 5). 

### 3.3. Post-Test Analysis

A representative SEM micrograph of a Ni-CYSZ anode surface and EDS spectra taken from the selected area after the long-term stability tests in wet and dry CH_4_ at 850 °C are presented in Figure 8. The anode had a porous surface and the darker contrasts in the amplified image of the selected area can be associated with the presence of carbon, as confirmed by EDS analysis. It can be seen that carbon particles are not homogeneously distributed on the surface. It is also worth noting that traces of carbon nanofibers were not found on any part of the analyzed surface. 

EDS analysis on the anode surface indicates an average concentration of carbon of ~18%, but the concentration varied between locations from 14% to 25%. EDS analysis only provides semi-quantitative results because the atomic weights of the components (e.g., C and metals) of the tested anode are very different. The fact that the amount of deposited carbon was small is consistent with the low performance drop observed during the stability test.

The micrographs of the anode/electrolyte interface (Figure 9) show that the structural integrity of the cell was maintained and delamination and/or micro-cracks were not detected after testing. The EDS mappings of Ni, Ce, Zr, Y, and C are also presented in Figure 9b–f. As expected, a homogeneous distribution of the common elements (Zr and Y) along the anode/electrolyte interface was observed. Cerium is well dispersed in the anode, while Ni appears as small particles that are also well distributed in this component. The presence of C is also evidenced in the mapping. It can be seen that carbon deposits are mainly located near to Ni particles, while the rest of the anode remains a C-free surface. The presence of ceria cannot avoid the formation of C, but it can inhibit its accumulation by promoting the electrochemical oxidation of carbon deposits.

The partial coverage of the Ni surface by C leads to a reduction in the TPB. In addition, the deposited carbon blocked the anode pores and hindered gas transportation, thus causing an enhancement of the diffusion over potential of the anode. All of these factors led to a continuous decrease in the cell performance. Although carbon is present on the Ni-CYSZ anode, the cell kept working and neither severe structural damage nor electrical failure were detected during testing, probably because carbon fibers were not formed in the anode. Although filamentous carbon can theoretically be formed below 650 °C, the presence of fibers in the Ni-YSZ cermet operated in CH_4_ at high temperature (800–900 °C) has been reported in the literature by several authors [4,5,6,7,8,9,10]. It was proposed that the formation of carbon fibers at these high temperatures is probably due to the dissolution of adsorbed carbon in the metal crystallite, followed by its diffusion through the metal and precipitation as carbon fibers on the surface at the rear of the metal particle. Therefore, the absence of carbon fibers in the Ni-CYSZ anode could also be attributed to the presence of Ce, which could enhance the oxygen storage and lead to an increase in the methane oxidation rate. In addition, ceria has a mixed ionic-electronic conductivity that expands the reaction zone of the anode beyond the TPB. 

## 4. Conclusions

A nickel-ceria-yttria stabilized zirconia (Ni-CYSZ) cermet material has been investigated as the anode material for the direct utilization of methane in SOFCs at 850 °C. Initially, the electrochemical behavior of the cell was evaluated in wet CH_4_ (3% H_2_O) by varying the load demand from 100 to 200 mA·cm^–2^ during 114 h, with each load applied for 24 h. Except for 200 mA·cm^–2^, the cell performance increased with load demand and reached the highest value at 180 mA·cm^–2^. All impedance spectra measurements showed three arcs, namely a high (HF), a medium (MF), and a low frequency (LF) arc, and these were assigned to the charge-transfer process, the adsorption/desorption of charged species or surface diffusion of the adsorbed species and non-charge processes (oxygen surface exchange, gas diffusion and possible chemical reactions), respectively. Impedance results revealed that polarization resistance (R_p_ = R_HF_ + R_MF_ + R_LF_) decreased on increasing both the load demand and operation time, a change due to the increase in the amount of oxygen ions favoring the electrochemical reactions. However, at 200 mA·cm^–2^, an enhancement in R_p_ was observed, and this was caused by R_MF_, which indicates a higher adsorption of species on the anode surface.

Long-term tests were carried out by operating the cell under a constant load demand of 180 mA·cm^−2^ at 850 °C in wet and dry CH_4_ for 236 h and 526 h, respectively. Similar cell performance was observed in wet (3% H_2_O) and dry CH_4_ during 236 h. The presence of 3% H_2_O in the feed is therefore not relevant when the cell is operated under a load demand of 180 mA·cm^-2^. The cell exhibited a low degradation rate of ~0.2 mV·h^–1^ during a total of 806 h operation under load demand. Analysis of impedance data revealed an increase in R_HF_ with operating time, probably caused by the formation on the anode of carbon deposits that cover the anode surface and also block the anode pores, thus reducing their connectivity. 

Post-material analysis by SEM and EDS showed carbon deposits mainly located at the top of the anode section and associated with Ni particles. Clear evidence of carbon deposits on cerium particles was not found. Therefore, the incorporation of cerium in the anode composition could reduce the formation of carbon deposits and improve the cell lifetime.

## Figures and Tables

**Figure 1 materials-13-00599-f001:**
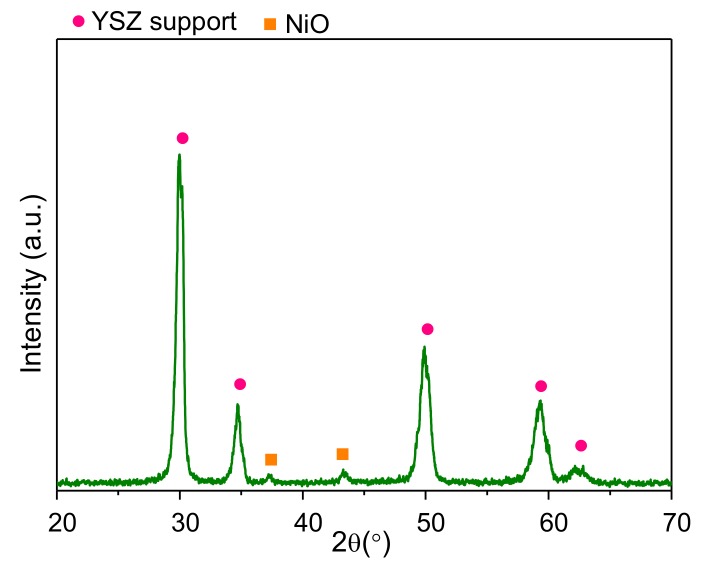
XRD pattern of the Ni-CYSZ material. (●) Tetragonal corresponding to YSZ supports and (■) NiO.

**Figure 2 materials-13-00599-f002:**
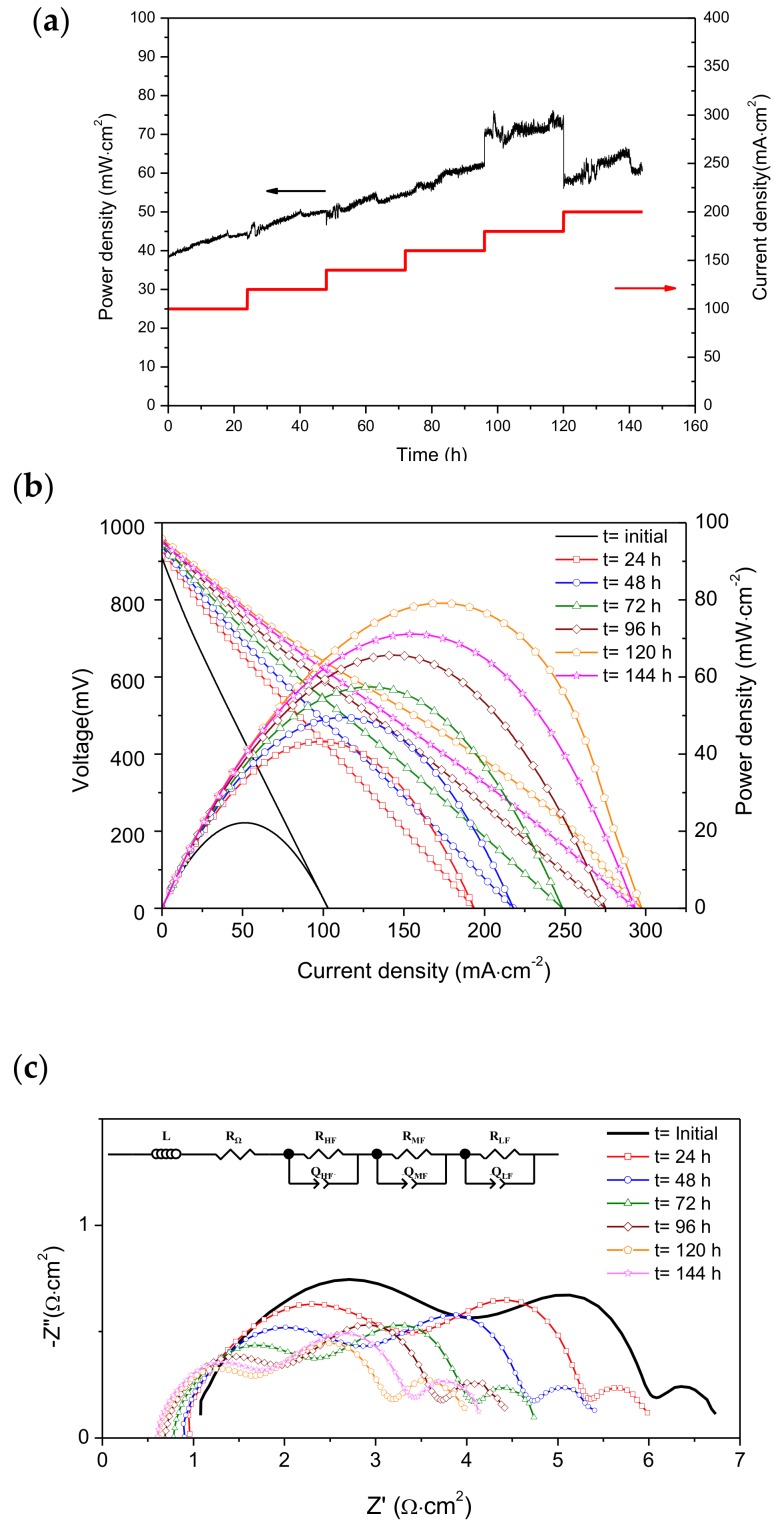
(**a**) Results of electrochemical tests on the cell (Ni-CYSZ/YSZ/LSM) in wet CH_4_ (3% H_2_O) at 850 °C. Power density and current density versus time, (**b**) I-V curves, and (**c**) impedance spectra (the inset shows the equivalent circuit model used for impedance spectra fitting).

**Figure 3 materials-13-00599-f003:**
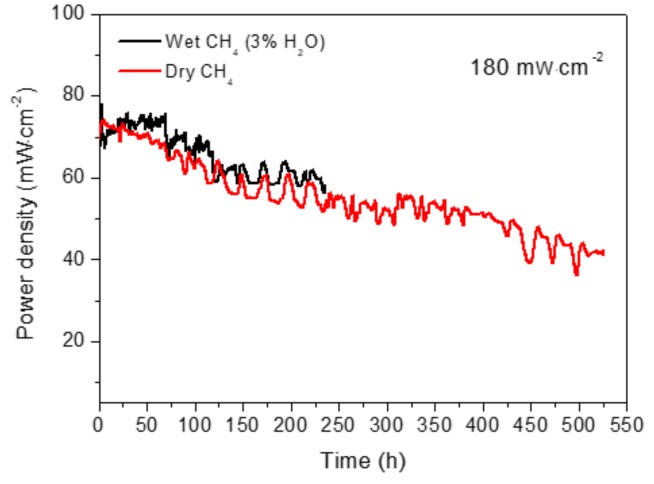
Long-term stability of the cell (Ni-CYSZ/YSZ/LSM) operating under a constant current density of 180 mA·cm^−2^ in wet (3% H_2_O) and dry CH_4_ at 850 °C.

**Figure 4 materials-13-00599-f004:**
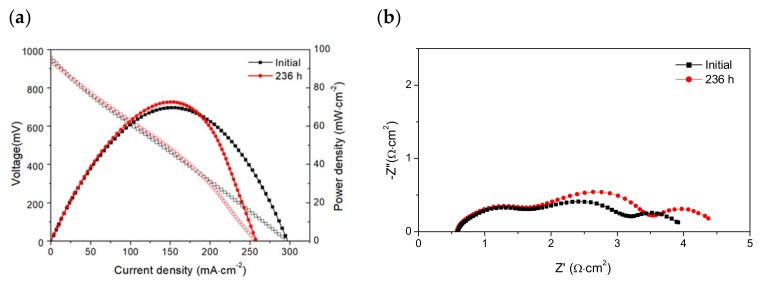
(**a**) I-V curves and (**b**) impedance spectra of the cell (Ni-CYSZ/YSZ/LSM) at 850 °C in wet CH_4_ (3% H_2_O) before and after the long-term stability test (236 h).

**Figure 5 materials-13-00599-f005:**
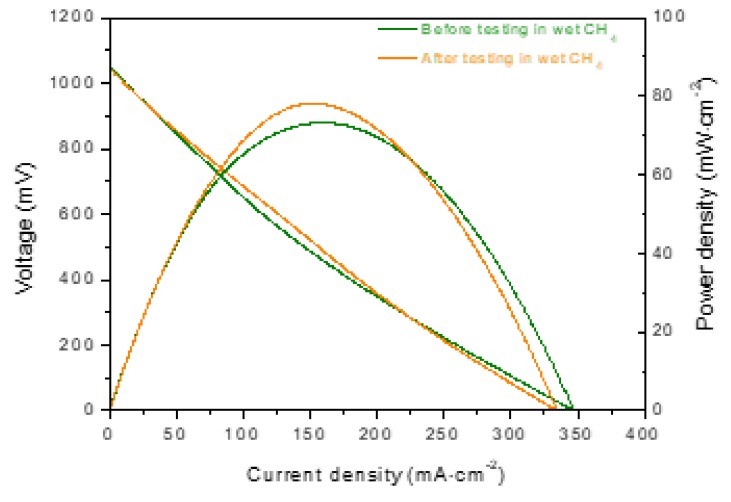
I-V curves of the cell (Ni-CYSZ/YSZ/LSM) at 850 °C in H_2_ (3% H_2_O) before and after long-term stability tests in wet CH_4_ (3% H_2_O).

**Figure 6 materials-13-00599-f006:**
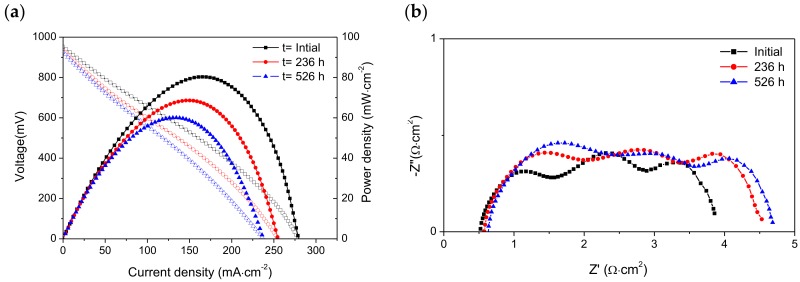
(**a**) I-V curves and (**b**) impedance spectra of the cell (Ni-CYSZ/YSZ/LSM) at 850 °C in dry CH_4_ before and after long-term stability testing for 236 h and 526 h.

**Figure 7 materials-13-00599-f007:**
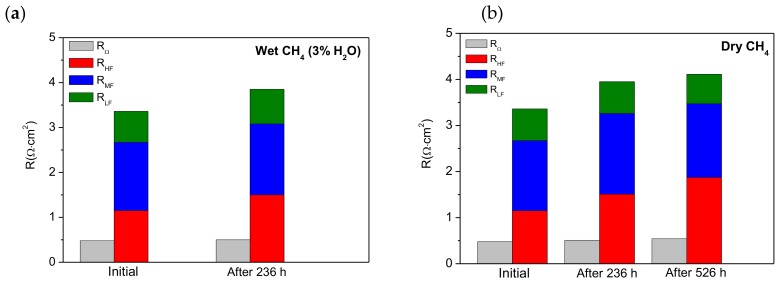
The R_Ω_, R_HF_, R_MF_, and R_LF_ values obtained from the impedance fitting to the equivalent circuit measured for the cell at 850 °C: (**a**) before and after 236 h operation in wet CH_4_ (3% H_2_O) and (**b**) before and after 236 and 526 h operation in dry CH_4_.

**Figure 8 materials-13-00599-f008:**
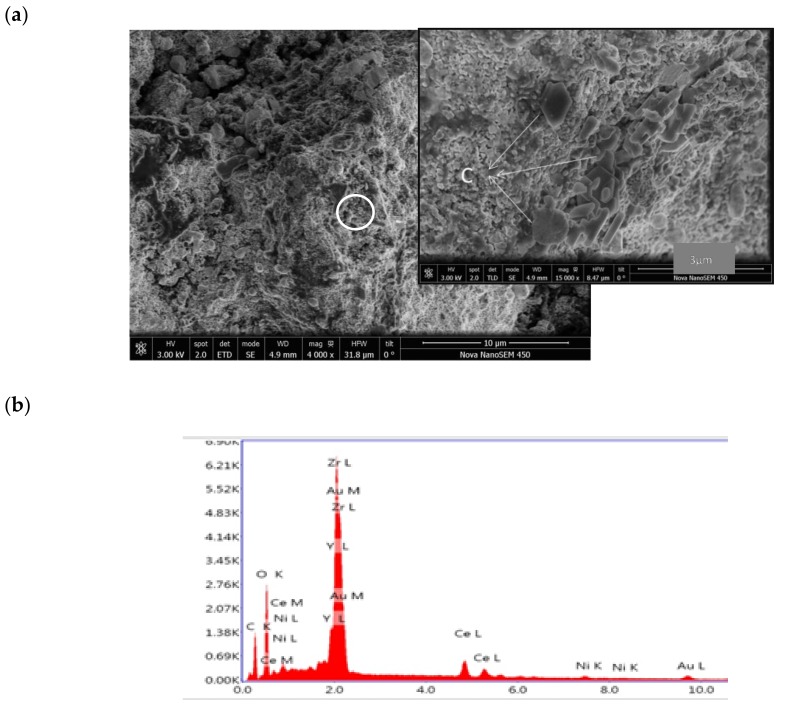
(**a**) SEM micrographs of the anode surface and (**b**) EDS pattern of a selected area of the anode surface after long-term tests in wet (3% H_2_O) and dry CH_4_ at 850 °C.

**Figure 9 materials-13-00599-f009:**
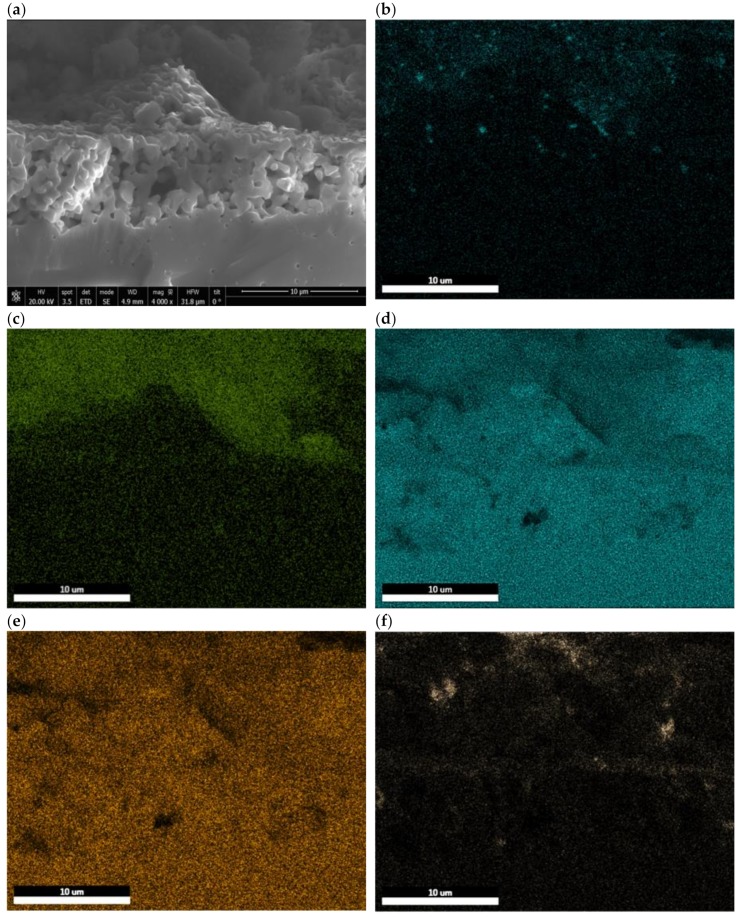
(**a**) SEM micrographs of the anode/electrolyte interface and elemental mapping of the anode/electrolyte interface (**b**) Ni, (**c**) Ce, (**d**) Zr, (**e**) Y, and (**f**) C after long-term tests in wet and dry CH_4_ at 850 °C.

**Table 1 materials-13-00599-t001:** Resistance and capacitance values obtained from the impedance data fitting shown in Figure 2c to equivalent circuit under wet CH_4_ (3% H_2_O) for different operating times at 850 °C.

Operating Time (h)	R_Ω_ (Ω·cm^2^)	High Frequency Arc		Medium Frequency Arc		Low Frequency Arc	
		R_HF_ (Ω·cm^2^)	C_HF_ (F·cm^–2^)	R_MF_ (Ω·cm^2^)	C_MF_ (F·cm^–2^)	R_LF_ (Ω·cm^2^)	C_LF_ (F·cm^–2^)
Initial	0.89	3.38	7 × 10^–6^	1.86	5.7 10^–3^	0.58	5.2
24	0.79	2.38	9 10^–6^	1.81	6.3 10^–3^	0.55	6.1
48	0.76	2.34	11 10^–6^	1.67	6.7 10^–3^	0.58	5.2
72	0.67	1.94	15 10^–6^	1.52	7.5 10^–3^	0.59	5.4
96	0.59	1.67	24 10^–6^	1.51	8.5 10^–3^	0.64	5.2
120	0.54	1.43	34 10^–6^	1.28	11.2 10^–3^	0.70	4.6
144	0.52	1.54	34 10^–6^	1.41	10.9 10^–3^	0.65	5.1

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
