# Peer review of "Performance of a Direct Methane Solid Oxide Fuel Cell Using Nickel-Ceria-Yttria Stabilized Zirconia as the Anode"

_materials, 2020, doi:10.3390/ma13030599_

Round 1
Reviewer 1 Report
The authors presented new important data on catalytic properties of nickel-ceria-yttria stabilized zirconia (Ni-CYSZ) cermet material as anode for direct oxidation of methane in a SOFC. The manuscript can be accepted for publication after some major and some minor corrections, listed below:
General comment: the results presented in this article must be compared with the result for just nickel-yttria stabilized zirconia (Ni-YSZ) cermet anode prepared in the same way and measured in the same conditions!
Abstract:
“…potentiostatic…” instead of “…potenciostatic…” “…After operating, material analysis …” instead of “…After operating, a post material analysis …”Introduction:
“…It is well-known that Ni-YSZ catalyzes...” instead of “…It is well-known that Ni-YSZ catalysts...” “…presence of carbon fibers at high temperature...” instead of “…presence carbon fibers at high temperate...” “…its diffusion through...” instead of “…its diffusion trough...” “…back of the metal...” instead of “…back the metal...”Experimental:
The authors wrote that they used platinum and gold pastes. Please add the text where it will be described how exactly these pastes were used! It is important because the pastes can influence the catalytic properties of the electrodes!Results and discussion:
“…Ni-CYSZ material...” instead of “…Ni-CYSZ catalyst...” The authors wrote that “The Ce loading was 15% molar” and that “ceria is integrated in the support structure”, based on XRD study. But how the YSZ crystal structure can accommodate this big amount of cerium cations? Probably, ceria was not integrated in the YSZ structure? The authors need more evidence if they want to insist that ceria was integrated in the YSZ structure! The authors wrote that “lower OCV value under wet CH4 could be directly related to the dilution of fuel”. But what is OCV? It is an open-circuit voltage, also known as the electromotive force (emf), which is the maximum potential difference when there is no current and the circuit is not closed. And for YSZ electrolyte this emf is defined by the oxygen pressure difference between fuel and air sides. So, when the fuel is wet, it has more oxygen, due to partial decomposition of water at high temperatures. Therefore, the resulting OCV is lower! The authors wrote that “Both experimental OCV values are lower than theoretical values for a complete oxidation of methane (~1.05V) and a partial oxidation of methane (~1.25V)”. Again, there is an impression that the authors did not analyze well enough what exactly is OCV! OCV results from oxygen pressure difference between fuel and air sides, when there is no current and the circuit is not closed! It means that OCV value is proportionally low as oxygen pressure in fuel side is low. As simple as that. In OCV conditions (no current and the circuit is not closed), the oxygen cannot go from air side to fuel side. It can only diffuse between sides if the sealing is not good enough. So if we assume that the sealing is complete, the only oxygen that can react with methane is the oxygen that already is present in methane (and added by water introduction). And under flowing methane condition an equilibrium installs between all those reactions (1-11) in the presence of nickel-ceria-yttria stabilized zirconia (Ni-CYSZ) cermet material. That is why the OCV value has nothing to do with complete oxidation of methane or partial oxidation of methane. There is no oxygen enough to oxidize methane significantly! Only when the circuit is closed and there is oxygen current through the electrolyte, there may be enough oxygen to complete those reactions! But again, these are no longer OCV conditions! “…Current density...” instead of “…Current deposition...” “…a gradual increase...” instead of “…a gradually increase...” “…originate a surplus...” instead of “…originate be a surplus...” “…cannot be removed by...” instead of “…cannot remove by...” “…spectra were also fitted...” instead of “…spectra was also fitted...” “…change oxygen stoichiometry...” instead of “…change of oxygen nonstoichiometric...” “…Ni-CYSZ...” instead of “…Ni-C-YSZ...” twice.Figure 1 caption:
“…XRD pattern of Ni-CYSZ material...” instead of “…XRD pattern of Ni-CYSZ catalysts...”Author Response

Reviewer 2 Report
This study is “Performance of a Direct Methane Solid Oxide Fuel Cell Using Nickel-Ceria-Yttria Stabilized Zirconia as Anode”. The topic is very interesting for reader. However, it should be minor revised according to the following comments:
The presentation of English is not good. The abstract and conclusions are trivial and not sound. The authors should add the novelty and worthy of this study in abstract and conclusions sections. In page 6, Lines 212-217 have been deleted. This paragraph is incomplete. The authors should rewrite this paragraph. In Fig. 6 (b), the authors should redraw the vertical axis from 0 to 1 to enlarge the differences of three curves.Author Response
Please see the attachment.

Round 2
Reviewer 1 Report
The authors improved significantly the manuscript, so it can be published.